# OpenReview forum: "FORCE: Transferable Visual Jailbreaking Attacks via Feature Over-Reliance CorrEction"
_ICLR.cc/2026/Conference — ICLR 2026 Conference Withdrawn Submission_

### Official Review · Reviewer_Gebp · 2025-10-27

**Soundness:** 3
**Presentation:** 4
**Contribution:** 3
**Rating:** 4
**Confidence:** 3

**Summary:**

The paper focuses on transferable visual jailbreaking attacks. It provides a detailed analysis from the perspectives of loss landscape, feature representation, and frequency-domain features to explain why current optimization-based attacks suffer from poor transferability. To address this issue, the authors propose a new approach, Feature Over-Reliance Correction (FORCE), which enhances the transferability of adversarial attacks.

**Strengths:**

1. The focus on transferable visual jailbreaking addresses an important topic in the safety community.


2. The paper provides a thorough analysis of why optimization-based attacks exhibit poor transferability and generalization.


3. The paper is clearly written and easy to follow.


4. The proposed method improves transferability, with especially strong gains on adapter-based MLLMs.

**Weaknesses:**

1. The proposed method shows clear improvement on adapter-based MLLMs, but for early-fusion and commercial MLLMs, although it performs slightly better than the baseline, the results are still far from being practically useful.


2. There are no experiments on defenses, which makes it difficult to assess the robustness of the proposed attack.


3. Transferable adversarial attacks have been well-studied in traditional CV and NLP domains. However, the discussion of these prior approaches, especially regarding their applicability to MLLMs, is missing from this paper.

**Questions:**

1. As the authors analyzed, current adversarial attacks are highly sensitive to weight perturbations. A straightforward idea would be to introduce weight perturbations during the optimization process. Do the authors have any insights on this?


2. In the proposed method, the authors choose to enlarge the feature representation region, which I think I understand the motivation behind. However, an opposite idea, finding adversarial examples whose neighboring regions in the feature space are more similar, might also reduce sensitivity. I’m curious how the authors view this opposite alternative.


Overall, the paper is well-motivated. If the authors can adequately address the concerns mentioned above, I would be inclined to raise my rating.

---

> ### Author Response · Authors · 2025-11-14
> **Response to Reviewer Gebp (1/2)**
>
> We sincerely appreciate your time and effort in reviewing our manuscript. Below are our responses to your valuable comments.
>
> ---
>
>
> >**Q1: Performance on Early-Fusion and commercial MLLMs**
>
> A1:
> The adapter-based and early-fusion MLLMs employ fundamentally different strategies for encoding images.
> Adapter-based models (e.g., CLIP-based architectures) operate directly in a differentiable pixel space, whereas early-fusion models construct a non-differentiable, tokenised representation of image patches.
> This discrepancy leads to significant challenges in achieving transferable visual jailbreaking attacks, as vulnerabilities that exist in tokenised spaces may map to non-integer or physically implausible pixel patterns, which lack meaningful interpretations in the real world.
> Consequently, although millions of jailbreaking examples may be feasible in theory, only a small subset corresponds to physically meaningful solutions.
> We therefore argue that while the absolute improvement from our method is around 3%, the relative improvement is close to 100%, which indicates that our approach effectively removes nearly half of the infeasible examples, which constitutes a non-trivial contribution to reducing the feasible solution space.
>
> Moreover, our method is designed to provide a deeper understanding of the transferability problem, and therefore, we intentionally exclude performance-oriented strategies such as prompt modification, model ensembling, or loss-based selection.
> Finally, we acknowledge that current research on visual jailbreaking is still at an early stage.
> However, the continued strengthening of textual alignment and the growing practical importance of multimodal evaluation suggest that visual jailbreak attacks will become an increasingly promising and valuable direction for future study.
>
>
> ---
>
> >**Q2: Performance Under Defence Method**
>
> A2:
> One advantage of optimisation-based jailbreaking attacks is their stealthiness, as they build on human-imperceptible perturbations rather than adversarial text suffixes.
> Moreover, our method does not depend on maliciously crafted semantic images; it functions effectively even with a blank initialisation (RGB (128, 128, 128)), making it highly difficult to detect.
> The only defence that may have an impact is injecting random noise into the image.
> However, applying strong noise levels would also degrade the model’s utility on benign tasks.
> We will include this experiment in the future version.
>
> For text-input detection, our method specifically targets visual jailbreaking, so we keep the textual prompts plain to ensure that the observed effects originate from the visual modality.
> Nevertheless, our method can be integrated into existing prompt-modification approaches to avoid text-based detection.
>
> ---
> >**Q3: Traditional CV and NLP Transferability**
>
> A3:
> We will discuss traditional CV and NLP transferability in the future version. However, in our view, most existing methods cannot be directly applied to our setting.
> This is because:
> (1) they aim to improve transferability in classification tasks, which differ substantially from generative tasks in both problem formulation and difficulty;
> (2) they focus on misleading models on benign tasks (i.e., tasks that are inherently harmless), which are not constrained by safety guardrails, whereas our method aims to elicit harmful knowledge; and
> (3) they have the knowledge of the target object (either an image or text), allowing the construction of a correspondence between the source and target. In contrast, our setting does not have knowledge of the intended harmful target content (e.g., the detailed instructions for building a bomb) for our supervision signal.

---

> ### Author Response · Authors · 2025-11-14
> **Response to Reviewer Gebp (2/2)**
>
> >**Q4: Weight Perturbation For Defence**
>
> A4:
> Prior work on Sharpness-Aware Minimization (SAM) and Adversarial Weight Perturbation (AWP) has shown that applying weight perturbations during training can improve model generalisation and robustness by encouraging a flatter loss landscape within the training perturbation radius. In general, incorporating weight perturbation can indeed yield a more robust model, especially against black-box attacks.
> However, from the attacker’s perspective, a white-box model can still be jailbroken with sufficient attack budget, because weight perturbation only encourages flatness within a limited perturbation radius. Using an infinitely large radius is infeasible, as it would collapse the model.
> Moreover, generative models encompass a much broader range of input–output mappings and conceptual spaces than classification models. Training with weight perturbations requires a target loss signal derived from a specific harmful reply, and such supervision is inherently limited in its ability to cover the full conceptual space.
> This also explains why adversarial training is often less effective than preference-optimisation methods in generative settings.
>
> ---
> >**Q5: Invariant feature points**
>
> A5:
> At the beginning of this project, we attempted to map neighbouring regions in the input space to invariant feature points at each subsequent layer by directly minimising the feature distance in Eq. 6. This approach indeed performs better than standard PGD in most test cases.
> However, this method also introduces instability. It essentially enforces a point-to-point mapping across layers. When the model parameters change during transfer, if the invariant point is disrupted at any layer, the downstream features may diverge significantly from the target output.
> In contrast, our current method encourages larger feasible feature regions, which can be viewed as a region-to-region mapping. This provides greater resilience to parameter changes and reduces the likelihood of failure during transfer.
>
> ---
>
> Thank you once again for your time and thoughtful review. We will incorporate your feedback and improve our work in future versions.

---

### Official Review · Reviewer_WwTk · 2025-10-30

**Soundness:** 2
**Presentation:** 2
**Contribution:** 3
**Rating:** 4
**Confidence:** 4

**Summary:**

This paper investigates transferable adversarial attacks on MLLMs, first examining their effectiveness in representation space and then proposing a solution.

**Strengths:**

1. The paper focuses on an important research question
2. The paper proposes an intriguing method and validates its performance on different benchmarks.

**Weaknesses:**

1. Clarity of writing (Section 3.1 Motivation).
The exposition is unclear and significantly hinders comprehension. In Section 3.1 (motivation), please clarify:
(a) What is the purpose of adding random noise?
(b) Why does adding adversarial noise increase the optimization loss? If the objective is to elicit “Sure, here it is,” this seems contradictory.
(c) In the bottom figure, what is the purpose of perturbing the model weights? In addition, the claim that “the attack is trapped in a local optimum of the source MLLM” is not evident from the visualization. The figures should be redrawn and more clearly explained to support the stated motivation. This ambiguity also propagates to later sections (e.g., Sec. 3.2).

2. Interpretation of Figure 3.
The explanation provided is insufficient. In Figure 3, layers 1, 6, 16, and 21 exhibit almost identical feature sensitivity, i.e., the required interpolation rate is around 0.2 for all. This observation does not convincingly support the claim that lower layers have worse generalization. Please reconcile this discrepancy or provide additional evidence/analysis.

3. Experimental setup and baselines.
(a) All noises are generated using LLaVA-1.5-7B, lacking cross-model validation across different open-source MLLMs.
(b) The method is not compared against other transferable MLLM adversarial attacks. such as [1,2], which is necessary to contextualize performance and transferability.

[1] A Frustratingly Simple Yet Highly Effective Attack Baseline: Over 90% Success Rate Against the Strong Black-box Models of GPT-4.5/4o/o1 NeurIPS 2025

[2] M-Attack-V2: Pushing the Frontier of Black-Box LVLM Attacks via Fine-Grained Detail Targeting. Arxiv 2025

**Questions:**

Please first refer to the weakness section.

---

> ### Author Response · Authors · 2025-11-14
> **Response to Reviewer WwTk**
>
> We sincerely appreciate your time and effort in reviewing our manuscript. Below are our responses to your valuable comments.
>
> ---
>
>
> >**Q1: Clarity of writing**
>
> A1:
> (a) Adding random noise is a standard technique for visualising a 3D loss landscape.
>
> (b) We apologise for the lack of clarity. In our method, the jailbreaking attack aims to minimise the loss toward the target response (“Sure, here is”). In this visualisation, the origin point (0,0) corresponds to the optimised and successful jailbreaking example. Then, the adversarial perturbation is applied along the gradient-ascent direction (increasing the loss) to illustrate the sharpness of the loss surface around the optimised jailbreaking attack.
>
> (c) Our claim that the attack becomes trapped in a local optimum of the source MLLM arises from two factors, as the loss jointly depends on both the jailbreaking perturbation and the model parameters. Therefore, we also visualise the loss landscape with respect to the model parameters to more comprehensively support this perspective.
>
> (d) The landscape may appear visually flat despite being intrinsically sharp because we use an extremely small noise magnitude. In classification settings, noise levels around 0.5 are common, whereas here we apply only 0.0005, as the jailbreaking attack is extremely fragile within the local minimum. Using 0.5 would indeed produce a very sharp landscape, but it would no longer be informative, since the attack already fails within the first 0.0005.
>
>
> ---
>
> >**Q2: Feature Sensitivity**
>
> A2:
> In Figure 3, although the interpolation rate at which failure occurs is the same, the subsequent loss trajectories differ substantially across layers.
> The early layers exhibit more non-linearity than the later layers, indicating that their feature landscapes are significantly more shapely and sensitive compared to later layers.
> Moreover, we provide an additional experiment in Appendix A that interpolates the feature between two different successful jailbreaking examples.
> These results show that, under interpolation, six points fail in Layers 1, 6, and 11, whereas only two points fail in Layer 16, and none of the points fail in Layer 21.
> These observations further support our claim that early layers possess much narrower feasible regions.
>
> ---
> >**Q3: Compare with the different models and SOTA baseline**
>
> A3:
> (a) We will report the attack results obtained using different models for generation in the future version.
>
> (b) We would like to point out that our work is one of the first studies to discover the limited transferability of optimisation-based visual jailbreak attacks identified in [1] (ICLR 2025),  and existing methods cannot be directly applied to this setting.
> For example, first, M-Attack [2,3] focuses on misleading the model on benign tasks (e.g., image description), which is not constrained by safety guardrails, whereas our method aims to elicit harmful knowledge.
> Second, M-Attack have the knowledge of the target object (the image needs to be described), allowing the method to construct a relationship between the source and target images. In contrast, our setting does not have knowledge of the intended harmful target content (e.g., the detailed instructions for building a bomb) for our supervision signal.
> To our knowledge, directly comparable baselines under the same setting are still relatively limited.
> But, our method can be simply plugged into generative-based visual jailbreaking pipelines, where a semantically corresponding image is first synthesised, and our optimisation strategy is subsequently applied to improve performance, as illustrated in the HADES results in our paper.
> In the future version, we will include comparisons with optimisation-based textual jailbreaking attacks, such as GCG.
>
> ---
>
> Thank you once again for your time and thoughtful review. We will incorporate your feedback and improve our work in future versions.
>
> >**Reference**
>
> [1] Failures to find transferable image jailbreaks between vision-language models. ICLR. 2025.\
> [2] A Frustratingly Simple Yet Highly Effective Attack Baseline: Over 90% Success Rate Against the Strong Black-box Models of GPT-4.5/4o/o1 NeurIPS 2025\
> [3] M-Attack-V2: Pushing the Frontier of Black-Box LVLM Attacks via Fine-Grained Detail Targeting. Arxiv 2025

---

### Official Review · Reviewer_9Z2k · 2025-10-31

**Soundness:** 3
**Presentation:** 3
**Contribution:** 3
**Rating:** 4
**Confidence:** 4

**Summary:**

This paper proposes the **Feature Over-Reliance CorrEction (FORCE)** method to enhance the cross-model transferability of visual jailbreaking attacks on multimodal large language models (MLLMs). The method addresses the issue of attacks relying on model-specific features by incorporating layer-aware regularization and spectral rescaling. Extensive experiments demonstrate that FORCE significantly improves the transferability and efficiency of visual attacks across various MLLM architectures, including commercial models.

**Strengths:**

1. The paper offers a clear and well-supported analysis of the underlying reasons for poor transferability in visual jailbreaking attacks. By examining the loss landscape of these attacks, the authors identify that attacks often reside in high-sharpness regions, making them highly sensitive to small changes in model parameters.

2. FORCE effectively addresses the problem of over-reliance on model-specific features in visual jailbreaking attacks. The method combines layer-aware regularization and spectral rescaling to improve the generalizability of the attacks across different models.

3. The experiments conducted demonstrate the effectiveness of FORCE in improving attack transferability across various MLLM architectures. The method is shown to outperform baseline approaches, achieving higher attack success rates and reducing the number of queries required for successful attacks.

**Weaknesses:**

1. The paper compares FORCE mainly with standard PGD and a few basic methods, but it does not evaluate the proposed method against more recent state-of-the-art transferable adversarial attack techniques.

2. The experiments do not assess how FORCE-generated attacks perform when models are equipped with defense mechanisms.

**Questions:**

1. The paper should provide concrete measurements of the computational cost and memory footprint for the proposed FORCE method, as the additional operations for layer regularization and spectral rescaling likely impose significant overhead.

2. The number of baselines compared in your work is relatively limited. Several existing white-box attack methods, including textual attacks like GCG, offer transferable attack strategies by learning a universal suffix for transferability. It would be beneficial to understand why your approach does not include comparisons with these methods.

3. It is important to evaluate how FORCE performs against defended models, particularly testing its effectiveness when target models employ common input-level defenses

---

> ### Author Response · Authors · 2025-11-14
> **Response to Reviewer 9Z2k**
>
> We sincerely appreciate your time and effort in reviewing our manuscript. Below are our responses to your valuable comments.
>
> ---
> >**Q1: Compare with the SOTA baseline**
>
> A1:
> We would like to point out that our work is one of the first studies to discover the limited transferability of optimisation-based visual jailbreak attacks identified in [1] (ICLR 2025).
> To our knowledge, directly comparable baselines under the same setting are still relatively limited.
> But, our method can be simply plugged into generative-based visual jailbreaking pipelines, where a semantically corresponding image is first synthesised, and our optimisation strategy is subsequently applied to improve performance, as illustrated in the HADES results in our paper.
> In the future version, we will include comparisons with optimisation-based textual jailbreaking attacks, such as GCG.
>
> ---
>
> >**Q2: Performance Under Defence Method**
>
> A2:
> One advantage of optimisation-based jailbreaking attacks is their stealthiness, as they build on human-imperceptible perturbations rather than adversarial text suffixes.
> Moreover, our method does not depend on maliciously crafted semantic images; it functions effectively even with a blank initialisation (RGB (128, 128, 128)), making it highly difficult to detect.
> The only defence that may have an impact is injecting random noise into the image.
> However, applying strong noise levels would also degrade the model’s utility on benign tasks.
> We will include this experiment in the future version.
>
> For text-input detection, our method specifically targets visual jailbreaking, so we keep the textual prompts plain to ensure that the observed effects originate from the visual modality.
> Nevertheless, our method can be integrated into existing prompt-modification approaches to avoid text-based detection.
>
>
> >**Q3: Computational Cost**
>
> A3:
> We will report the computational cost and memory footprint in the future version.
>
> ---
>
>
> Thank you once again for your time and thoughtful review. We will incorporate your feedback and improve our work in future versions.
>
> >**Reference**
>
> [1] Failures to find transferable image jailbreaks between vision-language models. ICLR. 2025.

---

### Official Review · Reviewer_t3iQ · 2025-10-31

**Soundness:** 3
**Presentation:** 3
**Contribution:** 3
**Rating:** 6
**Confidence:** 4

**Summary:**

This paper investigates the reason of optimization-based visual jailbreaking attacks on multimodal large language models (MLLMs) show poor cross-model transferability. Through detailed analyses of the loss landscape, layer-wise feature representations, and spectral dependencies, the authors find that existing attacks over-rely on model-specific shallow features and semantically weak high-frequency components, leading to sharp and non-transferable adversarial regions. Extensive experiments across both open-source and commercial MLLMs (e.g., LLaVA, InstructBLIP, Qwen, Claude, Gemini, GPT-5) demonstrate consistent and substantial improvements in cross-model transferability.

**Strengths:**

1. The paper provides a clear and multi-perspective diagnosis of the transferability issue, combining loss landscape visualization, layer interpolation, and Fourier analysis. And the proposed method seems practically effective in improving cross-model robustness of visual jailbreaks.

2. The method is tested on a wide variety of MLLMs (adapter-based, early-fusion, and commercial) and three benchmark datasets, showing consistent ASR gains (up to 20%) and reduced query costs. Component analysis, frequency influence plots, and feature interpolation experiments convincingly support the claimed mechanisms.

3. The paper is well-written and easy to follow.

**Weaknesses:**

1. The weighting rule w_m = \min(1, \ell_{m-1}/\ell_m) is somewhat ad-hoc and not derived from optimization principles; an adaptive or learning-based variant could be more convincing.

2. While the proposed FORCE demonstrates clear gains over PGD, the evaluation omits comparisons with other state-of-the-art transferable attack methods. Including these would strengthen the empirical evidence and contextualize the contribution.

3. “In this challenging setting, our method substantially improves transferability, achieving nearly a 100% increase over the baseline ASR”, corresponds to the ASR increase from 1% to 2%. I still think the performance can be limited. While I think the authors can briefly discuss why Early-Fusion MLLMs achieved limited attack performance.

**Questions:**

See weaknesses.

**Details Of Ethics Concerns:**

The paper focuses solely on improving attacks, without exploring how defenders might detect or mitigate such transferable jailbreaks. I suggest the author to briefly discuss how to mitigate the jailbreaks.

---

> ### Author Response · Authors · 2025-11-14
> **Response to Reviewer t3iQ**
>
> We sincerely appreciate your time and effort in reviewing our manuscript. Below are our responses to your valuable comments.
>
> ---
> >**Q1: The Design of Frequency Weight**
>
> A1:
> In this work, we observe that during the optimisation of jailbreaking attacks, high-frequency components increasingly and abnormally dominate the model’s behaviour, despite being largely model-specific and less semantically meaningful.
> Therefore, we down-weight the influence of high-frequency information according to the dynamic evolution of its relative relationship with lower-frequency bands throughout optimisation.
> This design principle follows from the intrinsic properties of natural images, where low-frequency yet semantically rich information exerts a stronger influence on both human and model visual perception than high-frequency, semantically weak features.
> Importantly, the frequency re-weighting process is inherently adaptive, as its strength is adjusted based on the loss computed at the current optimisation state.
>
> ---
>
> >**Q2: Compare with the SOTA baseline**
>
> A2:
> We would like to point out that our work is one of the first studies to discover the limited transferability of optimisation-based visual jailbreak attacks identified in [1] (ICLR 2025).
> To our knowledge, directly comparable baselines under the same setting are still relatively limited.
> But, our method can be simply plugged into generative-based visual jailbreaking pipelines, where a semantically corresponding image is first synthesised, and our optimisation strategy is subsequently applied to improve performance, as illustrated in the HADES results in our paper.
> In the future version, we will include comparisons with optimisation-based textual jailbreaking attacks, such as GCG.
>
> ---
>
> >**Q3: Performance on Early-Fusion MLLMs**
>
> A3:
> The adapter-based and early-fusion MLLMs employ fundamentally different strategies for encoding images.
> Adapter-based models (e.g., CLIP-based architectures) operate directly in a differentiable pixel space, whereas early-fusion models construct a non-differentiable, tokenised representation of image patches.
> This discrepancy leads to significant challenges in achieving transferable visual jailbreaking attacks, as vulnerabilities that exist in tokenised spaces may map to non-integer or physically implausible pixel patterns, which lack meaningful interpretations in the real world.
> Consequently, although millions of jailbreaking examples may be feasible in theory, only a small subset corresponds to physically meaningful solutions.
> We therefore argue that while the absolute improvement from our method is around 3%, the relative improvement is close to 100%, which indicates that our approach effectively removes nearly half of the infeasible examples, which constitutes a non-trivial contribution to reducing the feasible solution space.
>
> Moreover, our method is designed to provide a deeper understanding of the transferability problem, and therefore, we intentionally exclude performance-oriented strategies such as prompt modification, model ensembling, or loss-based selection.
> Finally, we acknowledge that current research on visual jailbreaking is still at an early stage.
> However, the continued strengthening of textual alignment and the growing practical importance of multimodal evaluation suggest that visual jailbreak attacks will become an increasingly promising and valuable direction for future study.
>
> ---
>
>
> Thank you once again for your time and thoughtful review. We will incorporate your feedback and improve our work in future versions.

---

### Author Response · Authors · 2025-11-14
**General Response**

Dear Reviewers,

We sincerely thank all reviewers for their thorough and constructive feedback. We are pleased that you found our work to address an important research question, offer clear and underlying insights into transferability, provide an effective method, andour paper is well-written and easy to follow.

We would like to highlight that our study is one of the first studies to identify the inherently limited transferability of optimisation-based visual jailbreaking attacks. Our work makes a non-trivial contribution by both deepening the understanding of the underlying causes and improving transfer performance. Since our goal is to gain principled insights into the transferability problem, we intentionally exclude performance-oriented techniques such as prompt modification, model ensembling, or loss-based selection.

We also acknowledge that research on visual jailbreaking remains at an early stage. Nevertheless, the continued strengthening of textual alignment and the increasing practical importance of multimodal evaluation suggest that visual jailbreak attacks will become an important and promising direction for future investigation.

Finally, we would like to express our gratitude once again. Your comments have substantially improved the clarity and quality of our manuscript. We will include additional results, covering more baselines, different optimised source models, defences, and computational cost, in the revised version.

Best regards,\
Authors

---

### Note · Authors · 2025-11-14

I have read and agree with the venue's withdrawal policy on behalf of myself and my co-authors.